# Acoustic Detector of Road Vehicles Based on Sound Intensity

**DOI:** 10.3390/s21237781

**Published:** 2021-11-23

**Authors:** Grzegorz Szwoch, Józef Kotus

**Affiliations:** Department of Multimedia Systems, Faculty of Electronics, Telecommunication and Informatics, Gdańsk University of Technology, 80-233 Gdańsk, Poland; jozef.kotus@pg.edu.pl

**Keywords:** vehicle detection, traffic monitoring, sound intensity, acoustic sensors

## Abstract

A method of detecting and counting road vehicles using an acoustic sensor placed by the road is presented. The sensor measures sound intensity in two directions: parallel and perpendicular to the road. The sound intensity analysis performs acoustic event detection. A normalized position of the sound source is tracked and used to determine if the detected event is related to a moving vehicle and to establish the direction of movement. The algorithm was tested on a continuous 24-h recording made in real-world conditions. The overall results were: recall 0.95, precision 0.95, F-score 0.95. In the analysis of one-hour slots, the worst results obtained in dense traffic were: recall 0.9, precision 0.93, F-score 0.91. The proposed method is intended for application in a network of traffic monitoring sensors, such as a smart city system. Its advantages include using a small, low cost and passive sensor, low algorithm complexity, and satisfactory detection accuracy.

## 1. Introduction

Traffic monitoring is an important element of modern smart city systems [1]. A distributed network of remote sensors provides data on traffic intensity within the observed area. The choice of a sensor type used for the traffic monitoring must balance the accuracy of vehicle detection with the cost of sensors and energy consumption. The sensors should also not interfere with the environment, and they should be robust to external interference.

There are several sensor types that may be considered for a smart city traffic monitoring system [2,3,4], but there are some issues with their practical implementation. Inductive loops belong to the most accurate vehicle detection sensors, but they must be mounted in the road surface, which is not always a viable solution. Pneumatic tubes are used only for ad-hoc measurements; they cannot be permanently mounted because they obstruct the traffic. Another group of possible solutions includes sensors that are positioned by the side of the road, or over the road. Radar sensors are the most often used devices belonging to this group [2]. They provide a satisfactory level of detection accuracy, but they are susceptible to electromagnetic interference (e.g., from power lines or cell base stations), and they emit electromagnetic waves, which may interfere with other equipment and increase the level of electromagnetic pollution in the environment. It may be an important issue if a large number of sensors must be installed in the area. Lidar sensors are a modern alternative to the radar sensors. They are efficient in vehicle detection, but they are susceptible to lighting conditions and they are more expensive than radars. The accuracy of both radar and lidar sensors degrades in a very dense traffic. In such scenarios, vehicle detection methods based on video analysis may be used [5,6]. Images from video cameras are analyzed using deep learning algorithms that detect individual vehicles in the image [7]. This approach is effective in dense traffic, but the analysis requires extensive computations and expensive equipment.

Acoustic sensors are not the mainstream of the state-of-art traffic monitoring solutions, but they provide an interesting alternative to the most often used ones. Such sensors do not emit any signals, they rely on the analysis of sound waves emitted by road vehicles, mostly from the tires and the engine. Several approaches to vehicle detection and counting with acoustic sensors may be found in the literature. Many of these methods are based on the analysis of signals recorded with two microphones placed by the road. Duffner et al. used a cross-power spectrum algorithm [8]. López-Valcarce et al. employed a maximum likelihood approach [9]. Ishida et al. used dynamic time warping [10] and sound mapping [11,12]. Uchino et al. [13] and Ishida et al. [14] evaluated the time difference of arrival method, with wind noise suppression. Kubo et al. used another method based on the discrete wavelet transform [15]. An alternative approach to vehicle detection is employing an array of acoustic sensors. Na et al. used an array of 37 microphones for detection of vehicle positions on multiple lanes [16]. Toyoda et al. proposed a method based on the peak detection of the power envelopes in signals obtained from an ad-hoc microphone array [17]. Marmaroli et al. compared the theoretical and the measured correlation time series using the two-dimensional Bravais–Pearson correlation coefficient [18]. Other approaches to acoustic vehicle detection include methods based on machine learning, e.g., by Gatto et al. [19], or utilizing field-programmable analog arrays (Bhattacharyya et al. [20]). Both the dual-microphone setups and the microphone arrays have large size, which may be an issue in practical applications.

The research described in this paper aimed to propose an efficient vehicle detection method based on an acoustic sensor, which is suitable for a large-scale smart city system and does not have the issues of the existing sensors, described earlier. The main difference between the proposed approach and the methods found in the literature is that our method is based on the analysis of sound intensity signals measured in a two-dimensional space, which allows for determining the incoming sound direction. To our knowledge, sound intensity analysis was not used for traffic monitoring in any work published earlier. Our initial research in this area was described in the previous publication [21] in which we implemented a simplified vehicle detector for the traffic intensity estimation. That detector worked well for isolated vehicles, but it was too inaccurate in more intensive traffic. In another publication [22], we described the average traffic speed estimation by the sound intensity analysis. However, the vehicle detection was not a part of that algorithm. In this paper, we propose a reworked vehicle detector that utilizes two-dimensional sound intensity measurement. This method aims to detect road vehicles (both the isolated ones and vehicles moving in a sequence) and to determine their direction of movement. This detector can also provide data needed by the average speed estimation algorithm [22].

The rest of the paper is organized as follows. First, we describe the algorithm from the intensity signals to the final detection result. Next, we validate the method using real-life recordings and reference data. We also present examples of vehicle detection in typical scenarios. The paper ends with a discussion of the obtained results and conclusions.

## 2. Materials and Methods

The algorithm may be summarized as follows (Figure 1). First, sound intensity signals, measured in two orthogonal directions, are smoothed, and the total intensity is calculated. The next stage is acoustic event detection. The total intensity is compared with the background noise estimate, and if it exceeds the detection threshold, an acoustic event is detected, otherwise, the background noise estimate is updated. Each detected event is then analyzed to check whether it represents a moving vehicle and to divide the event into individual vehicles, if necessary. For this task, a normalized source position is calculated and processed by a Kalman filter which estimates the position and the velocity of the source. The range of the estimated position changes allows for the detection of moving sound sources, and the velocity estimate is used to determine the direction of movement. The details of each stage are presented in the following subsections.

### 2.1. Sound Intensity Calculation and Preprocessing

The proposed method assumes that the sound intensity is measured with a sensor positioned by the observed road, and the coordinate system is oriented so that the X-axis is parallel to the road (directed right) and the Y-axis is perpendicular to the road, directed towards the road (Figure 2). If the axes of the sensor are oriented differently, the intensity signals must be converted by applying a rotation matrix. The sensor provides the sound intensity measurements along the X and Y axes, denoted *I_X_* and *I_Y_*.

Sound intensity is a vector quantity that describes the energy flow in sound waves, defined as the power carried by sound waves per unit area in a direction perpendicular to that area [23,24]. The proposed method does not rely on a specific method of measuring *I_X_* and *I_Y_*. The signals may be measured with a sound intensity probe, such as microflown [25], which provides the particle velocity and pressure signals. Sound particle velocity may also be measured with thermal sensors [26,27]. Our experiments used a custom sound intensity sensor built from pressure sensors (microphones), closely spaced in pairs on each axis (four microphones in total). An instantaneous sound intensity may be calculated from Euler’s equation, using a finite difference approximation. For a single axis, on which the pressure is measured with two microphones, *p*_1_(*t*) and *p*_2_(*t*), the instantaneous intensity at the middle point between the microphones is given by [24]:(1)$$I_{inst}\left(t\right)=-\frac{p_{1}\left(t\right)+p_{2}\left(t\right)}{2\rho \Delta r}\int_{-\infty }^{t}\left(p_{2}\left(t\right)-p_{1}\left(t\right)\right)\;dt$$
where *ρ* is the air density and Δ*r* is the spacing between the sensors. For the proposed algorithm, the constant term in the denominator may be dropped.

The actual sound intensity is calculated by integrating the instantaneous intensity over the period *T_avg_*:(2)$$I\left(t\right)=\int_{t-T_{avg}}^{t}I_{inst}\left(t\right)\;dt$$

As a result, a discrete-time intensity signal with the sampling period of *T_avg_*, is obtained.

From this point on, we will be using *I_X_* and *I_Y_* to denote the magnitude of the sound intensity vectors oriented along the axes of the coordinate system. A total intensity *I_XY_* on the *X*-*Y* plane is a scalar value given by:(3)$$I_{XY,n}=\sqrt{I_{X,n}^{2}+I_{Y,n}^{2}}$$
where *n* is the index of the discrete-time signal. The total intensity represents the energy of the sound wave measured on the *XY* plane.

The intensity signals after the integration usually contain a high level of noise which originates from both the sensors and the inaccuracy of the intensity calculation method. Therefore, noise suppression is required before the total intensity is calculated [22]. We used a simple moving average filter which proved to work very well in this case, providing the necessary amount of smoothing to the intensity signal. A delay is introduced in the process of filtering, but it does not influence further signal analysis. We also experimented with a Savitzky–Golay filter, but it preserved too much variation in the intensity signals, making the analysis more problematic.

Figure 3 shows the sound intensity plots (unprocessed and smoothed) calculated from a recording of two vehicles passing by the sensor. It should be noted that the sensor measures the sound intensity of an apparent sound source that is a superposition of all sound sources in a vehicle, and it does not represent a constant point in the vehicle body [22]. The value of *I_X_* is the largest when the apparent sound source is close to the sensor, and it decreases as the distance from the sensor increases, so the plot resembles an inverted parabola. The *I_Y_* value is zero when the source is on the *Y*-axis. When the source moves away from the sensor, *I_Y_* initially increases, then it starts to decrease because of attenuation of the sound wave energy with an increasing distance from the source. The total intensity *I_XY_* is almost equal to *I_X_* when the source is close to the sensor because this component is dominant. At larger distances, the *I_Y_* component influences the *I_XY_* value.

### 2.2. Acoustic Event Detection

An acoustic event indicates a presence of a sound source, either stationary or moving. If no sound source is present in the vicinity of the sensor, the intensity signals contain only background noise. Event detection is performed by comparing the current total intensity *I_XY_* with a background noise intensity estimate *I_BN_*. The condition of acoustic event detection is:(4)$$I_{XY,n}>I_{BN,n}+b$$
where the constant *b* is the detection margin.

The state of the event detector is active if condition (4) is fulfilled, otherwise it is inactive. The change of the state signals the beginning or the end of the event. If the detector is inactive (no sound source detected), the background noise estimate is updated:(5)$$I_{BN,n}=\alpha \;I_{BN,(n-1)}+\left(1-\alpha \right)I_{XY,(n-\tau )}$$
where α is the averaging factor, *τ* is the update delay. Introducing the delay is necessary, otherwise, the value of *I_BN_* would increase on the rising edge of *I_XY_* when the event begins.

When vehicles move close to each other in dense traffic, the events representing individual vehicles are merged into an event group. Therefore, it is necessary to divide the group into individual vehicles. If a sufficient spacing between the sound sources exists so that each event contains a peak in I*_XY_*, the following procedure is used. A sliding window of length *L* (where *L* is an odd number) is applied to *I_XY_* within the event. A peak at position *j* is detected if:(6)$$\arg\max_{j}I_{XY,j}=n-\frac{L-1}{2},\;j\in \langle n-L+1,n\rangle$$

Valleys are detected as local minima between the detected peaks. The group is then divided into events using the detected valleys as breaking points. The peak value *I*_max_ = *I_XY,j_* for each event is recorded.

Figure 4 presents an example of acoustic event detection. The first event represents a single vehicle. Next, a group of five vehicles moving close to each other in the same direction is detected. This group is divided into individual vehicles by finding local maxima and minima (peaks and valleys). The value *L* should be adjusted to the smallest expected spacing between the vehicles. The choice of optimal *L* is discussed further in the paper.

### 2.3. Source Position and Velocity

Once an acoustic event is detected, the next step is to determine whether it was caused by a moving vehicle. The main advantage of a method based on sound intensity, compared with standard pressure- or energy-based methods, is that it is able to track changes of the incoming sound direction. In the coordinate system shown in Figure 2, the azimuth *φ* of a sound source is calculated relative to the *Y*-axis (*φ* = 0 if the source is on the *Y*-axis):(7)$$\varphi =\arctan\left(\frac{I_{X}}{I_{Y}}\right)$$

If we denote the position of a sound source as (*x*, *y*), then:(8)$$\tan\varphi =\frac{x}{y}$$

Combining Equations (7) and (8), we obtain:(9)$$x=y\frac{I_{X}}{I_{Y}}$$

It is not possible to measure *y* with the sound intensity sensor. However, in the vehicle detection algorithm, we do not need to know the exact source position, we only need to track its changes. Therefore, we can set *y* = 1 and we obtain a normalized source position:(10)$$x_{n}=\frac{I_{X,n}}{I_{Y,n}}$$

A physical interpretation of the position signal is as follows: *x_n_* is a projection of a sound source position onto a trajectory parallel to the *X*-axis of the sensor, at *y* = 1. If an ideal point source was moving at a constant speed along this trajectory, the plot of its position signal would be a straight line.

For the analysis of the source movement, we require that the position signal is monotonic. However, *x_n_* calculated from Equation (10) is usually contaminated by noise which is mostly a result of variations of the real sound source position on both *X* and *Y* axes. Therefore, filtering the position signal is necessary. It is possible to use another moving average filter to smooth the position signal. However, the movement of a source is a kinematic process that may be modeled using a Kalman filter [28,29]. We assume a one-dimensional movement of the source projected onto a normalized trajectory *y =* 1, with a constant velocity *v*:(11)$$x_{n}=x_{0}+v\cdot n\cdot T_{S}$$
where *T_S_* is the sampling period of the discrete position signal. We denote the filter state as **x** = [*x_n_*, *v_n_*], where *x_n_* is the observed variable and *v_n_* is a hidden variable.

For each sample of the calculated position signal, two stages of Kalman filtering are performed: a prediction followed by an update. The prediction stage calculates a prior estimate of the filter state (the time step is equal to one sample):(12)$$\left[\begin{array}{c} {\bar{x}}_{n}\\ {\bar{v}}_{n} \end{array}\right]=\left[\begin{array}{cc} 1 & 1\\ 0 & 1 \end{array}\right]\cdot \left[\begin{array}{c} x_{n-1}\\ v_{n-1} \end{array}\right]$$

In the update stage, the estimates are calculated according to the measurement of *x_n_* (obtained from Equation (10)) and the noise covariance matrices that model the process noise (deviations from the constant velocity model) and the measurement noise (inaccuracy of the measured position). The exact Kalman filter equations are found in the literature [28,29] and they are not repeated here. The variance of both noise types is the most important parameter that influences the filtering result. The optimal choice of the variance values is a complex problem that lies outside the scope of this paper. In our case, the variance of the measurement noise must be significantly higher than the variance of the process noise to obtain a desired degree of the position signal smoothing.

The velocity estimate obtained from the Kalman filter is used to determine the direction of the source movement and to reject events that represent stationary sound sources (with a velocity close to zero). It should be noted that these estimates represent the movement of the source projected onto the normalized trajectory. It is not possible to measure the physical speed of a road vehicle because the distance y to the source is unknown, and due to the movement of the apparent sound source within the vehicle. These issues are discussed in more detail in our previous publication [22].

Figure 5 presents an example of processing the position signal with a Kalman filter for the intensity signals shown in Figure 3. A smooth, monotonic position function, suitable for further analysis, is obtained. For the first vehicle, moving left-to-right, the position increases towards positive values, while for the other vehicle, moving right-to-left, the position changes in the opposite direction. The direction of the source movement is determined by examining the sign of the source velocity (estimated by the Kalman filter) at the point of maximum intensity within the detected event.

Next, a vector **x** of filtered position values belonging to a single event is constructed:(13)$$x=\left\{x_{i}:i\in \langle n_{1},n_{2}\rangle \wedge \left|v_{i}\right|\leq v_{\max}\right\}$$
where *n*_1_ and *n*_2_ are the indices of the beginning and the end of the event, respectively, and *v*_max_ is used to exclude values with higher velocity than expected. Such a case may happen at the edges of the event if the source transitions from the previous position (e.g., with multiple vehicles in a sequence). A position span *s* of the event is then calculated as:(14)s=maxx−minx

For an isolated moving vehicle, it is expected that the span *s* covers the observed area of the road. For a sequence of vehicles, the span of the whole sequence should also cover the observed range, but the individual events may have a limited range of position changes, depending on the distance between the vehicles.

### 2.4. The Decision Stage

Finally, the results obtained from the previous analysis stages are used to establish whether the detected event represents a moving vehicle. At this point, the detected events are described using a set of parameters: event duration *N* in samples, minimum and maximum position (*x*_min_, *x*_max_), position span *s*, maximum intensity value *I*_max_, velocity *v* (measured at the maximum event intensity).

In some cases, an event related to a single vehicle may exhibit multiple peaks in the intensity signal. A truck with a trailer often produces such an effect. The procedure described earlier divides this event into two separate events, which would cause a false positive detection. Therefore, the decision module checks whether the minimum and maximum positions for two adjacent events **x**_1_ and **x**_2_ fulfill the condition:(15)$$x_{1}{}_{\min}<x_{2}{}_{\max}\;\wedge \;x_{1\max}>x_{2\min}$$

If the two events belong to the same vehicle, the position range of the second event will continue the range of the first event. However, if the two events belong to separate vehicles, there will be a transition of the source position between the vehicles (a movement in the opposite direction), so the position ranges of the two events will overlap.

The remaining part of the decision module performs testing the event parameters against the threshold values. The following conditions must all be fulfilled.

The event duration test: *N* > *N_thr_*. This test eliminates short-term events that are usually caused by impulsive noise sources.The maximum intensity test: *I*_max_ > *I_thr_*. Events with low sound intensity are discarded because they may not represent moving vehicles.The span test: *s* > *s_thr_*. It is expected that the movement of a vehicle is observed in a sufficiently wide range of positions. For a group of connected events (multiple vehicles moving close to each other), the span *s* should be calculated for the whole group, as the individual vehicles may have a limited span.

An event that passes all the tests is assumed to represent a moving vehicle observed in a sufficient range of position changes, and with a sufficient sound intensity. The movement direction is denoted as ‘1’ if *v* > 0 and ‘−1’ if *v* < 0. This concludes the detection procedure.

## 3. Experiments

### 3.1. Test Setup

Performance of the proposed method was examined using signals recorded in real-world conditions. We have reused the dataset recorded for the purpose of the experiments described in our previous publication [21]. The test setup, consisting of a sound intensity sensor and a microcomputer, was installed by a road in a suburban area (geographic coordinates: 54.344555, 18.443811). A section of a straight road with a speed limit of 90 km/h, with one lane in each direction, was observed by the sensor positioned 4 m away from the road edge, 2.9 m above the ground, oriented as shown in Figure 2. The signals were recorded continuously for 24 h (Monday, 14:00 to Tuesday, 14:00, local time). The temperature during the recording was 15 °C to 26 °C, average pressure was 997 h Pa, the wind was up to 7 m/s from the west, and there were occasional periods of light rainfall (about 15% of the total time).

The reference data were obtained using a certified vehicle counter system Metrocount MC5600, based on pneumatic tubes that were mounted on the road near the acoustic sensor. For each detected vehicle, a timestamp, the direction of movement, the vehicle speed and the vehicle wheelbase were recorded. Wheels of vehicles passing through the tubes produced an impulsive “thump” noise that was visible in the unprocessed intensity plots. However, the procedure of smoothing the intensity signals with a moving average filter resulted in removing these spikes. Therefore, the presence of the tubes did not influence the performance of the algorithm.

A custom sound intensity sensor was constructed to record the intensity signals [30]. The sensor consisted of six digital, omnidirectional MEMS microphones (IvenSense INMP441), placed in pairs on three orthogonal axes (the vertical axis was not used in the experiments). The microphones in each pair were spaced by 10 mm and the midpoints of all pairs were at the same position. The sensor was placed in a rubber shield with a protective fur cap. A six-channel digital signal from the sensor (sampled at 48 kHz, 24-bit resolution) was recorded on a microcomputer through an I^2^S-USB interface. The analysis was performed offline. We also did preliminary tests with online analysis using a Raspberry Pi 4 microcomputer, running the presented algorithm implemented as Python scripts, and the system was able to work as intended, the computational power of the hardware was sufficient for the algorithm to work in online mode.

The experiments were performed on a desktop computer. The algorithm was implemented using Python scripting language, version 3.10. The preprocessing stage involved calculation of the intensity signals. Sound intensity in two directions was calculated according to Equation (1). A calibration procedure was used to equalize the frequency responses of the microphones and the delay between the microphone pairs [30]. The microphone signals were processed with a band-pass filter, 400 Hz–4 kHz. The purpose of the filtering is rejection of the environmental noise at low frequencies [22]. Most of the sound wave energy emitted from moving road vehicles is concentrated in the middle frequency range (500 Hz to 2 kHz) [31].

The instantaneous intensity signals were then integrated (Equation (2)). We choose the integration period *T_avg_* = 256 samples (5.33 ms) as a compromise between the temporal resolution of the analysis and the signal-to-noise ratio. The intensity signals *I_X_* and *I_Y_*, sampled at 187.5 Hz, were then processed by a moving average filter of length 51 samples (272 ms).

The actual vehicle detection was performed using the processing system shown in Figure 1. The event detection module used the following parameters: *b* = 10^−4^, *I_BN_*_,0_ = 2·10^−4^ (Equation (4)), *τ* = 100 Sa, *α* = 0.98 (Equation (5)), *L* = 121 Sa (Equation (6)). In the position tracking module, *v*_max_ = 0.04 (Equation (13)), and the noise variance parameters in the Kalman filter were set experimentally to provide a sufficient level of signal smoothing: process noise variance 1.1·10^−5^, measurement noise variance 69.4, initial position and velocity variance: 0.028. Parameters of the decision block: *N_thr_* = 100 Sa, *I_thr_* = 5·10^−4^, *s_thr_* = 0.2.

All the detections from the algorithm (sample index and vehicle direction) were recorded and then compared with the reference data. The 24-h recording was split into one-hour slots, and the performance metrics were calculated for each slot and in total. The number of vehicles in each time slot is presented in Figure 6. The overall traffic intensity was moderate, it was sparse during the nighttime (<40 vehicles per hour), but it became more intensive during the rush hours (up to 452 vehicles per hour). Therefore, the dataset represents a varying degree of traffic intensity and allows for algorithm performance evaluation in different scenarios.

### 3.2. Examples of Vehicle Detection

Four detection examples representing typical scenarios are presented in this section. Each case is illustrated with a figure consisting of a plot of the intensity *I_XY_* and a plot of the source position *x* obtained from the Kalman filter. Edges of the detected events are marked with vertical dashed lines. Positions of the intensity maxima are marked with plus signs. Finally, the detection results are marked in the position plot as pins, indicating the direction of movement (positive or negative). The plots present normalized (unitless) values of intensity and position. It is possible to measure the sound intensity in physical units, but it requires a calibration of the sensor, this was not needed in the presented algorithm. The position is normalized by projecting the source on a normalized trajectory (*y* = 1).

Figure 7 presents the simplest case of detecting isolated vehicles. The source position changes in a monotonic way throughout the whole event. The first vehicle moves left-to-right, the second one right-to-left. Between the vehicles, the apparent source position remains at position 5.0–7.5 (the source ‘jumps’ from the first to the second vehicle).

In Figure 8, a sequence of vehicles moving left-to-right, close to each other, is presented. The first three vehicles form a single group, divided into individual events by the algorithm. The range of position changes is smaller than in the previous example because the next vehicle ‘captures’ the source position before the previous vehicle reaches the end of the observed range. Transitions of the source position between the vehicles are visible as sections of decreasing position. The algorithm must differentiate the position changes related to the moving vehicle from these resulting from a transition between two successive vehicles.

Figure 9 presents the case of vehicles passing each other: the first vehicle moves left-to-right, the second (just after the first) moves right-to-left. These two events form a single group. In this case, the position of the source increases for the first vehicle, then it decreases for the second vehicle, without a transition between them. A source transition (a segment of increasing position) is visible between the second and the third vehicle. The algorithm must be able to determine whether the change in the direction of position changes is caused by a vehicle moving in the opposite direction, or by a source transition between two vehicles moving in the same direction. In the proposed algorithm, the event detection module determines sections related to the individual vehicles. The maximum intensity point indicates the velocity value used to detect the direction of source movement.

Figure 10 shows an example of a case in which vehicle detection was not possible. According to the reference data, three vehicles were observed. The first two were moving left to right, and they were detected correctly. However, the third vehicle was moving right to left, and it was observed approximately at the same time as the second vehicle. This is an example of occlusion: the third vehicle (moving on the further lane) is obscured by the second vehicle which moves closer to the sensor, so the third vehicle is ‘invisible’ to the sensor. Occlusions are a common problem for most of the sensors installed by the road, including radars and lidars. In the presented example, the third vehicle blended into the falling edge of the intensity signal from the second vehicle, and it could not be detected. A section of small position decrease can be observed after the second vehicle, but it is not sufficient to determine that it represents a vehicle (it could be a source transition after the vehicle has passed).

### 3.3. Parameter Tuning

The choice of the algorithm parameter values was made by examining the algorithm performance on a one-hour recording. We chose the time slot 15–16, as it contained the highest number of vehicles. We counted the number of correct detections (true positives, *TP*), the number of missed vehicles and vehicles with incorrect direction detected (false negatives, *FN*), and the number of events that do not represent the actual vehicles or detections with incorrect direction (false positives, *FP*). We evaluated the algorithm performance by calculating the standard metrics. A recall describes the percentage of vehicles that were detected with correct direction:(16)$$recall=\frac{TP}{TP+FN}$$

A precision metric describes the percentage of detections that represent the actual vehicles:(17)$$precision=\frac{TP}{TP+FP}$$

The F-score *F*_1_ is a combined performance measure:(18)$$F_{1}=2\frac{recall\cdot precision}{recall+precision}$$

Optimization of the algorithm parameters usually involves finding the desired balance between the recall and the precision, depending on which aspect is more important for the application (less FNs or less FPs). During the experiments, we found out that two parameters have the most significant influence on the algorithm performance, namely, the minimum intensity peak *I_thr_* in the decision module, and the event detection window length *L*. Figure 11 presents the plots of the performance metrics calculated for different values of these parameters. If *I_thr_* is set too low, spurious local maxima may be detected as events, resulting in FPs. If it is too large, events with low intensity may not be detected (FNs). For the window size, if *L* is too small, local maxima may cause a fragmentation of the event, leading to FPs. For example, a car with a trailer may be detected as two separate vehicles. If *L* is too large, two or more vehicles moving close to each other may be merged into a single event, leading to FNs. In both cases, we found the values for which the recall and the precision have similar values, and the F-score reaches the maximum (these values are marked with the vertical dashed lines in the plots), and we selected these values (*I_thr_* = 5·10^−4^, *L* = 121) as the optimum in the presented experiment. These values should be adjusted according to the signal-to-noise ratio and the distance between the sensor and the road in practical installations. Other algorithm parameters had much less significant influence on the algorithm performance in our experiment, they may be tuned if necessary.

### 3.4. Results

The performance metrics were calculated for the whole 24-h observation period and for each one-hour slot to test the algorithm performance in varying traffic intensity. The results are presented in Table 1. From 5905 observed vehicles, 95% were detected correctly, and all the performance metrics were also about 0.95. This confirms that the overall performance of the proposed algorithms is satisfactory. Analysis of the hourly results confirms the expectation that the algorithm performance deteriorates with increasing traffic intensity. During the night (<40 vehicles/h), no FNs are observed, some FPs occur due to the low intensity of the events. During the day, when the traffic increases, the metrics are lower. The lowest scores were obtained for the 17–18 slot, but none of them were below 0.9. The algorithm performance is slightly worse for the afternoon rush hours than in the morning, this is caused by a higher percentage of vehicles moving on the further lane in the afternoon (Figure 6), because the detection on the further lane (right-to-left) is more problematic due to occlusions.

There are three main causes of the observed FNs. The first type of FN is related to the events with insufficient intensity (below *I_thr_*), e.g., when a vehicle is coasting slowly, especially on the further lane. The second FN type is caused by the occlusions discussed earlier (Figure 10). The third FN type is caused by determining the incorrect vehicle direction, despite the correct event detection. Such errors happen in a dense traffic, in event groups consisting of multiple vehicles moving in opposite directions. In some cases, the transition of the source position between the vehicles cannot be distinguished from position changes caused by a vehicle movement, and the direction is determined incorrectly. The FP results were less common in the experiment, and they were caused either by incorrect direction determination or by detecting local intensity maxima as separate events.

Table 2 shows a comparison of four detection modes: detection of both directions (the same as in Table 1, correct detection of direction is required to count as TP), detection of a single direction only (left to right or right to left, the other direction is discarded from the analysis), and detection without determining the direction (each detected vehicle is counted as TP). From these results, it may be concluded that 96 of the FNs (30%) are of the type 3 (incorrect direction), and most of the remaining FNs are of the type 2 (occlusion), because the number of FNs is much higher for the further lane (direction right-to-left). Similarly, 96 of FPs (41%) are caused by the incorrect direction detected, and the numbers of FPs in both directions are similar. Performance of the detector, measured by the F-score, is similar for the first three scenarios, and it is slightly higher (by 0.01) if only the closer lane is analyzed. On the further lane, the recall drops to 0.93, but the accuracy decrease is not significant. Finally, if the detection of the vehicle direction is not needed, a slightly higher accuracy (F-score 0.97) is achieved.

### 3.5. Comparison with Other Methods

In our previous publication [21], we used a simplified vehicle detector based on the sound intensity to estimate traffic intensity. The algorithm detected zero crossings in the source position and then used the intensity to determine whether they were related to acoustic events. Performance of this algorithm, examined using the same dataset as in this paper (only the left-to-right direction was analyzed), was suboptimal in dense traffic, because some of the source position changes in a group of vehicles did not cross the zero, so they were not detected. The algorithm presented in this paper significantly improves the F-score (0.95 vs. 0.9) and the recall (0.95 vs. 0.87), with a smaller increase in the precision (0.96 vs. 0.93) over the earlier algorithm. For comparison, in [21] we also used a radar sensor with a custom analysis algorithm, achieving the recall 0.91, the precision 0.98 and the F-score 0.95. The algorithm presented here achieved the same F-score, better recall, but worse precision.

It is also interesting to compare the performance of the proposed algorithm with other approaches to vehicle detection and counting, utilizing acoustic sensors (Table 3). However, it should be noted that a direct comparison is impossible because each algorithm was tested in different conditions, usually on a smaller number of vehicles (i.e., small variation in traffic density) and not every algorithm was designed to determine the direction. Performance of our algorithm matches the best methods based on a dual microphone setup (Kubo et al. [15]) and a microphone array (Marmaroli et al. [18]). Most of the other algorithms achieved high precision but poor recall below 0.9. The method proposed by Gatto et al. [19] achieved a slightly higher F-score and a much higher recall than our method, but it employed a complex algorithm based on machine learning.

## 4. Conclusions

We presented an algorithm for detecting and counting road vehicles in signals obtained from an acoustic sensor positioned by the road. The novel aspect of this algorithm is that it is based on the sound intensity measured in two orthogonal directions (parallel and perpendicular to the road). Acoustic events are detected by the analysis of the sound intensity changes. Since the sensor is able to determine the direction of the incoming sound, the algorithm tracks the changes of the normalized sound source position, determines if the acoustic event is caused by a vehicle moving on the observed road, and detects the direction of movement.

In comparison with other acoustic vehicle detection methods, our algorithm uses signals recorded by a small sensor consisting of closely spaced microphones, while most of the related methods use either two microphones placed relatively far away from each other or a large array of microphones. Unlike the state-of-art detection sensors, the proposed method does not require a sensor permanently installed in the road surface (like an inductive loop) or a sensor that emits waves into the environment (such as radars or lidars). The processing algorithm is computationally efficient, especially compared with complex algorithms based on machine learning to analyze video or audio data.

We evaluated the algorithm performance in a test covering the consecutive 24 h, with 5905 observed vehicles. This way, we were able to validate the algorithm in varying traffic intensity. Overall, we achieved the precision, the recall, and the F-score all close to 0.95, and in the densest traffic (250–450 vehicles per hour), none of the metrics were below 0.9. Thus, by comparison with other acoustic-based methods and other state-of-art sensors positioned by the road, we may conclude that the proposed method provides a satisfactory level of detection accuracy.

The presented experiment allowed for validation of the algorithm and for establishing a baseline of the algorithm performance. One problem that can be addressed in future research is the reduction of errors related to incorrect detection of the movement direction. The problem of occlusions cannot be resolved by the algorithm, and it is expected that the performance degrades if there are multiple lanes in the same direction as the frequency of occlusion increases. In this case, overhead sensors should be used. Analysis of sound intensity from a sensor mounted over the road is possible, but this is a separate problem that lies outside this paper’s scope. In the presented experiment, we tested the algorithm in specific conditions. It is expected that if the traffic intensity increases above 500 vehicles per hour, the algorithm performance degrades, but the degree of this degradation needs to be evaluated in future experiments. Weather conditions and the level of environmental noise are also factors that need to be tested in the future. At this moment, based on the experimental results, we may conclude that the proposed algorithm is an interesting alternative to the existing vehicle detection and counting methods.

The proposed method is intended for application in a network of traffic monitoring stations, which is a part of a smart city system or an intelligent transportation system. Continuous measurement of traffic intensity and its variations at multiple locations within the observed area allows for efficient management of a traffic network. It provides the drivers with up-to-date information about traffic intensity and optimal routes. The detector may also be used to provide input data to the average speed estimation algorithm proposed in our previous publication [22]. It is also possible to use the detector for controlling the elements of a smart city system, for example, to adjust the intensity of street lighting according to the traffic intensity. To construct a large network of sensors continuously monitoring the traffic in real time, the detection method should utilize low-cost and small size sensors that preferably should not emit any signals to the environment, and the signal analysis procedures should be able to work in real-time on low-end hardware. The proposed method, together with a sound intensity sensor, fulfills these requirements.

## Figures and Tables

**Figure 1 sensors-21-07781-f001:**
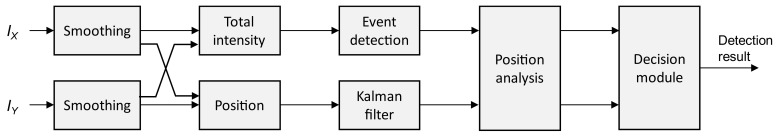
Block diagram of the proposed algorithm.

**Figure 2 sensors-21-07781-f002:**
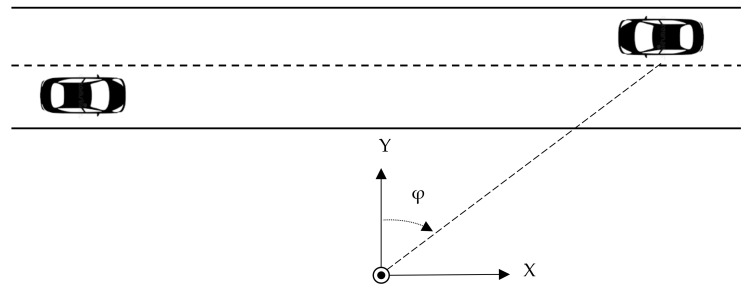
Orientation of the coordinate system relative to the road. A circle denotes the sensor position.

**Figure 3 sensors-21-07781-f003:**
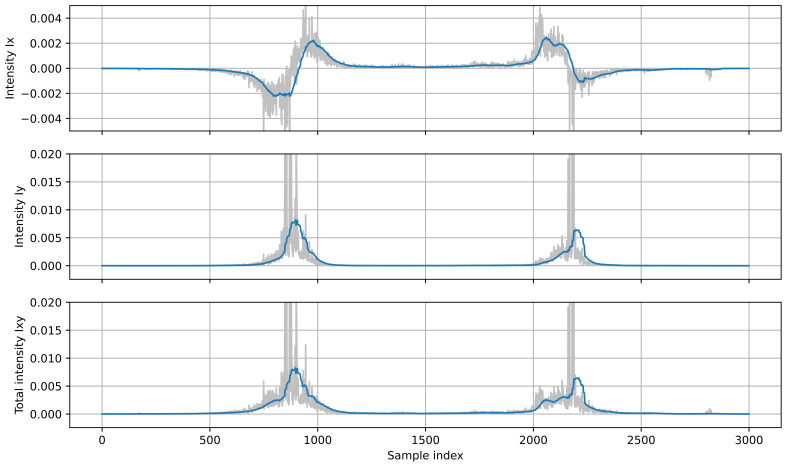
Plots of sound intensity: *I_X_*, *I_Y_* and total intensity *I_XY_*, calculated for two vehicles moving in opposite directions (first: left to right, second: right to left). Light lines show unprocessed signals, dark lines show filtered intensity signals.

**Figure 4 sensors-21-07781-f004:**
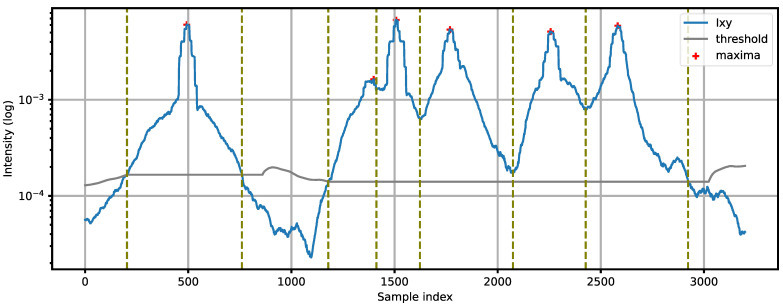
Example of acoustic event detection: a single vehicle followed by a group of five vehicles. Vertical dashed lines show the edges of the detected events.

**Figure 5 sensors-21-07781-f005:**
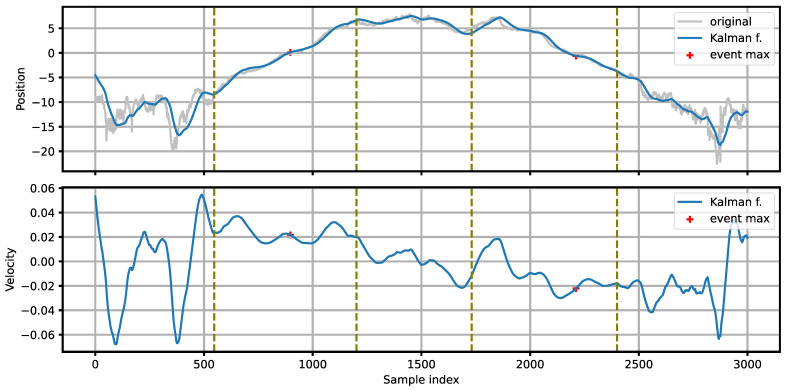
Position and velocity of the sound source, estimated with the Kalman filter. The light line in the position plot shows the input to the Kalman filter. The vertical dashed lines mark the edges of the detected events. The positions of the event maxima are marked with plus signs.

**Figure 6 sensors-21-07781-f006:**
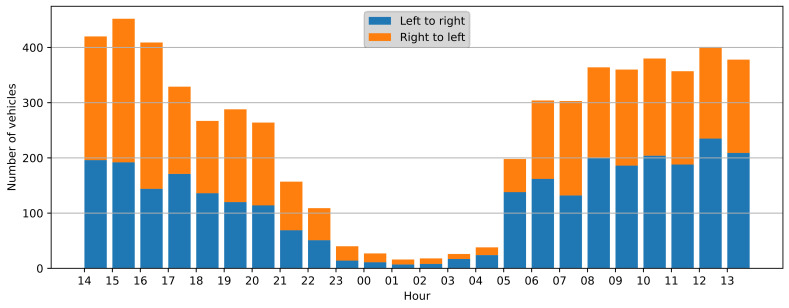
Number of vehicles observed in one-hour time slots.

**Figure 7 sensors-21-07781-f007:**
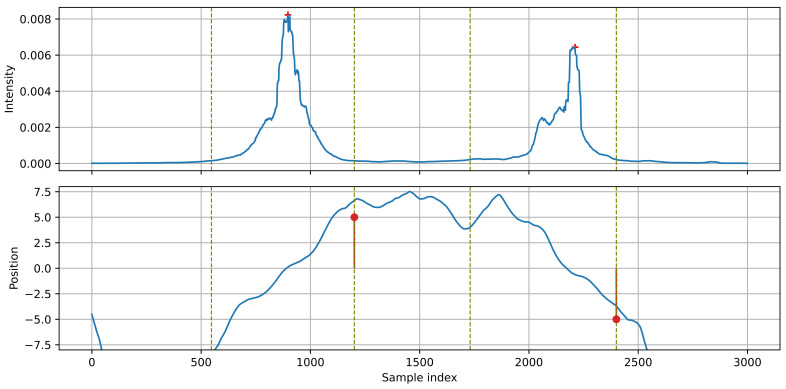
Example of detection: two isolated vehicles moving in opposite directions.

**Figure 8 sensors-21-07781-f008:**
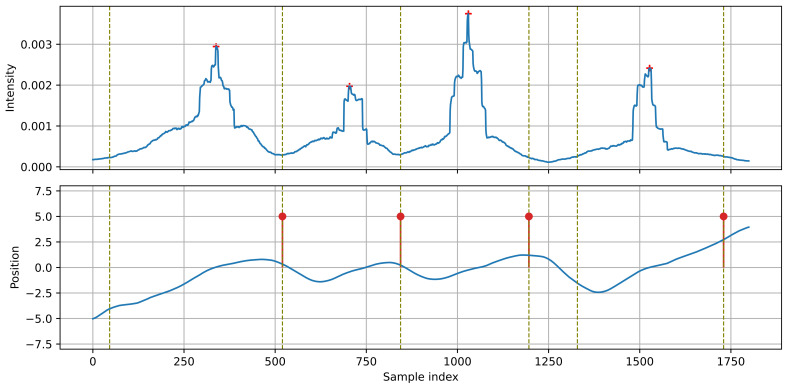
Example of detection: four vehicles moving close to each other, left to right.

**Figure 9 sensors-21-07781-f009:**
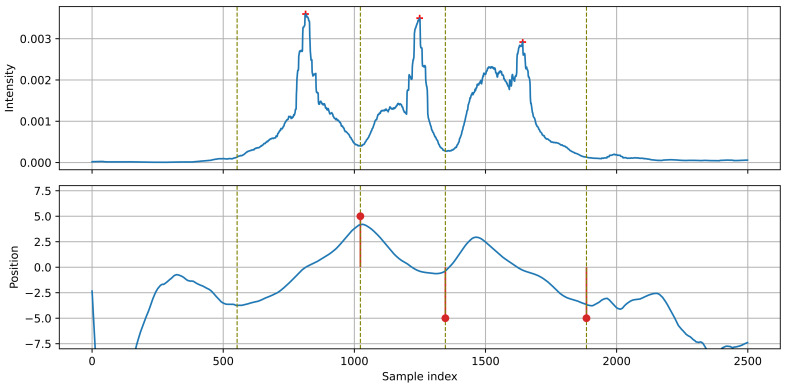
Example of detection: vehicles passing each other, moving in the opposite directions.

**Figure 10 sensors-21-07781-f010:**
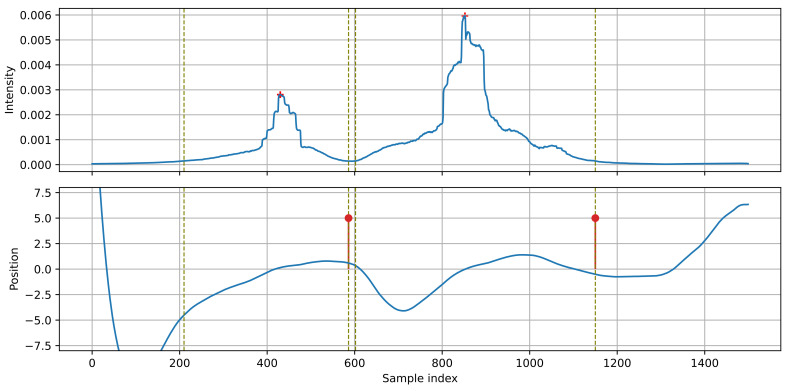
Example of an unsuccessful detection: the third vehicle is not detected.

**Figure 11 sensors-21-07781-f011:**
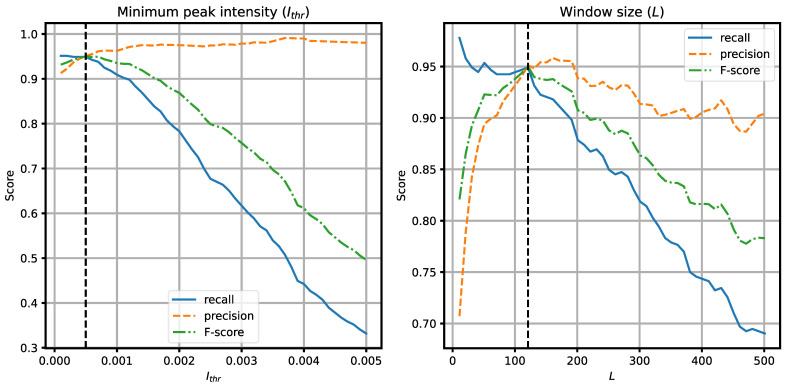
Optimization of the algorithm parameters: *N_thr_* and *L*. The vertical dashed lines indicate the values chosen for the experiments.

**Table 1 sensors-21-07781-t001:** Vehicle detection results: the total result for the 24-h observation and the results for one-hour time slots. *N* is the number of vehicles according to the reference data.

Hour	N	TP	FN	FP	Recall	Prec.	F-Score
Total	5905	5590	315	233	0.95	0.96	0.95
12–13	420	395	25	21	0.94	0.95	0.94
13–14	452	429	23	22	0.95	0.95	0.95
14–15	409	378	31	22	0.92	0.94	0.93
15–16	329	313	16	8	0.95	0.98	0.96
16–17	267	253	14	11	0.95	0.96	0.95
17–18	288	258	30	20	0.90	0.93	0.91
18–19	264	252	12	6	0.95	0.98	0.97
19–20	157	153	4	4	0.97	0.97	0.97
20–21	109	104	5	1	0.95	0.99	0.97
21–22	40	38	2	1	0.95	0.97	0.96
22–23	27	27	0	0	1.00	1.00	1.00
23–24	16	16	0	0	1.00	1.00	1.00
00–01	18	18	0	0	1.00	1.00	1.00
01–02	26	26	0	2	1.00	0.93	0.96
02–03	38	38	0	0	1.00	1.00	1.00
03–04	198	189	9	2	0.95	0.99	0.97
04–05	304	291	13	8	0.96	0.97	0.97
05–06	303	281	22	12	0.93	0.96	0.94
06–07	364	349	15	20	0.96	0.95	0.95
07–08	360	343	17	11	0.95	0.97	0.96
08–09	380	363	17	10	0.96	0.97	0.96
09–10	357	345	12	18	0.97	0.95	0.96
10–11	401	375	26	10	0.94	0.97	0.95
11–12	378	356	22	24	0.94	0.94	0.94

**Table 2 sensors-21-07781-t002:** Vehicle detection results in four scenarios: with detection of the direction (*Both*), for a single direction (left-to-right or right-to-left) and without detecting the direction (*No dir.*).

Dir.	N	TP	FN	FP	Recall	Prec.	F-Score
Both	5905	5590	315	233	0.95	0.96	0.95
L-R	2928	2818	110	119	0.96	0.96	0.96
R-L	2977	2772	205	114	0.93	0.96	0.95
No dir.	5905	5686	219	137	0.96	0.98	0.97

**Table 3 sensors-21-07781-t003:** A comparison of performance of vehicle detection algorithms based on acoustic sensors.

Algorithm	Sensor	N	Recall	Prec.	F-Score
The proposed method	intensity	5905	0.95	0.96	0.95
Czyżewski et al. [21]	intensity	2953	0.87	0.93	0.90
Duffner et al. [8]	dual microphones	1000+	0.81	N.A.	N.A.
Ishida et al. [10]	dual microphones	116	0.82	0.92	0.87
Ishida et al. [11]	dual microphones	176	0.85	1.00	0.92
Ishida et al. [12]	dual microphones	178	0.83	0.84	0.83
Uchino et al. [13]	dual microphones	133	0.80	0.75	0.77
Ishida et al. [14]	dual microphones	93	0.86	0.99	0.92
Kubo et al. [15]	dual microphones	151	0.95	0.94	0.95
Na et al. [16]	microphone array	1093	0.86	N.A.	N.A.
Toyoda et al. [17]	microphone array	64	0.89	0.89	0.89
Marmaroli et al. [18]	microphone array	139	0.94	0.97	0.95
Gatto et al. [19]	single microphone	2314	0.99	0.96	0.97

## Data Availability

Scripts and data used in the experiments are available in a git repository at https://git.pg.edu.pl/p16759/cardetpub (accessed on 23 November 2021).

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
