# Peer review of "Acoustic Detector of Road Vehicles Based on Sound Intensity"

_sensors, 2021, doi:10.3390/s21237781_

Round 1
Reviewer 1 Report
The author showed a thorough understanding of the current practice in traffic monitoring by presenting a varied set of solutions to this problem. This article suggested a simple method which requires two passive acoustic sensors to measure the traffic by examining sound intensity. It is satisfactory, in a sense that it achieved a good recall rate and precision. However, measuring traffic with acoustic technique is common. This paper didn't manage to achieve:
- A better recall rate and precision over existing method.
- more knowledge of the traffic such as coordinates of vehicles.
- better detection results when vehicles occlude each other.
In this sense, the researcher should dive deeper into its novelty.
That said, the paper presented a simple yet sufficient alternative method to measure the traffic, which is much appreciated in an industrial sense. Given the great effort the author yielded in the field test, I recommend that this paper be thoroughly revised.
Here are some suggestions:
- The abstract did not lay the ground for your study: What is insufficient of current practices and what will your work contribute to it?
- Line 84: sound … that … are the input data. -> better keep the top of your sentence light, as top-heavy sentences are difficult to read.
- Line 64, 117: an acoustic sensor or two sensors? -> Please check if there are other contradictions as well and correct them.
- Line 131, line 135 -> make sure to present your noise suppression method in a more detailed way.
- Line 163 -> Please showcase how far between two vehicles is sufficient enough to distinguish them.
Reviewer 2 Report
The authors present the application of an algorithm for detecting the presence and the direction of vehicles in urban scenario. The paper is well organized and well written and the
proposed technique is interesting. Moreover, each section is adequately discussed and presented.
However, I suggest the authors to address some points listed in the atteched the PDF file.

Round 2
Reviewer 1 Report
Dear Authors,
Thank you for submitting your responses for the first-round review. I appreciate the details you offered to address my questions about the novelty of your research as well as some minor issues. Your argument is strong and convincing, so I will suggest your paper be accepted.
Note that after revision you seem to have made quite a few typos or errors in spacing. Please check the following excerpts:
in considered for a smart... 26
bebelong to the most accurate... 28
First, Ssound intensity signals, 87
sensorsmicrophones, p1(t) and p2(t), the ... 121
Aa standard simple 136
without performance issuesthe com- 313
positivesnegatives, FPFN), and the number of events that do not represent the actual ve- 407
Author Response
We thank again the Reviewer for a thorough analysis of our manuscript and for suggesting that the manuscript is accepted. The typos that the Reviewer noticed are a result of reading the manuscript in a revision tracking mode. in which both the previous and the current text version are presented at the same time. It is our fault that we did not generate a PDF file ourselves, the autogenerated PDF retained the revision mode. We apologize for that. We rechecked the manuscript to make sure that the text is correct, we resubmit the same MS Word file again and we also upload the PDF file without revision tracking.